# The Effect of Supplementation of Rumen-Protected Choline on Reproductive and Productive Performances of Dairy Cows

**DOI:** 10.3390/ani12141807

**Published:** 2022-07-14

**Authors:** Indrė Mečionytė, Giedrius Palubinskas, Lina Anskienė, Renata Japertienė, Renalda Juodžentytė, Vytuolis Žilaitis

**Affiliations:** 1Department of Animal Breeding, Veterinary Academy, Lithuanian University of Health Sciences, Tilžės Str. 18, 47181 Kaunas, Lithuania; giedrius.palubinskas@lsmuni.lt (G.P.); lina.anskiene@lsmuni.lt (L.A.); renata.japertiene@lsmuni.lt (R.J.); renalda.juodzentyte@lsmuni.lt (R.J.); 2Large Animals Clinic, Veterinary Academy, Lithuanian University of Health Sciences, Tilžės Str. 18, 47181 Kaunas, Lithuania; vytuolis.zilaitis@lsmuni.lt

**Keywords:** RPC, BHB, progesterone, yielding, reproductive

## Abstract

**Simple Summary:**

Choline is a key methyl donor synthesized endogenously in mammals, and its availability is important for various biological functions. A review of the literature presents controversial results and deals with the question of whether rumen-protected choline supplemented during the transition period significantly affects cows’ health, production, and reproduction. However, most recent research shows that in dairy cattle, choline supplementation improves milk yield, composition, and fertility.

**Abstract:**

We aimed to evaluate the effects of organic herbal preparations containing rumen-protected choline (RPC) in dairy cow milk’s BHB and progesterone (P4) concentration changes, reproduction, and production performances. Cows were divided into the following two groups: The CHOL (*n* = 60) cow diet was supplemented with 10 g/day RPC from 20 days pre-calving to 20 days post-calving, and CONT (*n* = 60) were fed a conventional diet. BHB and P4 concentrations were measured at 5–64 DIM and 21–64 DIM, respectively, with DelPro 4.2. BHB was lower in the CHOL group at 5–64 DIM than CONT *p* > 0.05. The first post-calving P4 peak, *p* < 0.001, was determined earlier in the CHOL group, and the P4 profile during 21–64 DIM was similar, *p* > 0.05. The insemination rate was lower, and the interval between calvings was shorter. The first insemination time was earlier in the CHOL group, *p* < 0.05. Milk yield was higher in the CHOL group at 21–64 DIM, *p* > 0.05. The CHOL group had more fat in their milk at 31–60 DIM, *p* < 0.05. There were no significant differences in protein and SCC between the groups, *p* > 0.05. Based on our results, we concluded that the supplementation of RPC pre- and post-calving had statistically significant effects on the first peak of P4, and benefited the reproduction performances, milk yield, and milk fat during the early postpartum period.

## 1. Introduction

Increasing dairy cows’ productivity negatively impacts their health and reproduction [1]. This tendency is also characterized by high-yielding Lithuanian Black and White cows [2]. These cows belong to an old native dairy cattle breed with significant regional and cultural value, which accounts for 70% of all cattle in Lithuania. Therefore, this study and its results are relevant for researchers, farmers, and breeders of all high-yielding cows [3].

It has been reported that 40–70% of lactating dairy cows of different breeds and milk yield levels under different management systems develop metabolic and/or infectious diseases during the postpartum period [4]. The high milk yield is related to negative energy balance (NB), and endocrine and metabolic changes in postpartum cows. NB occurrence is associated with excessive fat mobilization in the form of non-esterified fatty acids (NEFAs). The phenomenon of NEFA mobilization continues with ketosis and fatty liver occurrence in dairy cows during the postpartum period. High NEFAs and ketones are negatively associated with cows’ health [5]. However, uterine pathologies such as subclinical endometritis (SE) play a primary role in the decline of reproductive performance serum levels of β-hydroxybutyrate acid (BHBA), albumin, and urea [6].

Most health disorders occur during the late gestation and early lactation transition periods [7]. The transition period is between 3 weeks pre-calving and 3–4 weeks post-calving. During that time, dairy cattle experience a negative energy balance (NEB) as a result of insufficient energy from low dry matter intake (DMI) [8,9]. NEB leads to lipolysis and the mobilization of adipose tissue reserves, which can cause productivity and fertility problems, and health disorders in dairy cows [10]. Bobe et al. [11] stated that the influx of NEFAs in the liver overwhelms the hepatic oxidation function. The growing energy demand on cows with high milk production was satisfied through lipid mobilization during PP. Triglycerides (TAG) are the main lipid molecules stored as an energy reserve in adipose tissue. The TAG molecule releases one glycerol and three non-esterified fatty acids (NEFAs) or long-chain non-esterified fatty acid molecules that are major energy sources for tissues during NEB periods. Some of the NEFAs entering the liver become hepatocytes. NEFAs are used for energy production through the Krebs cycle and can be re-esterified by triacylglycerols (TAG) and exported as very-low-density lipoproteins (VLDL) [12]. NEFAs’ role in the liver and peripheral tissues and their toxic effects in excess are explained and discussed in the literature [13].

Substantial lipid mobilization from adipose tissue increases oxidative stress, impairs immunity, and is associated with higher incidences of periparturient health problems [14].

The cow will likely experience fatty liver syndrome. The disturbance in hepatic function leads to a weakness in subsequent milk yield and is associated with poor reproductive success [15]. The prophylaxis of fatty liver and ketosis facilitates fatty acid metabolism in the liver, oxidation in mitochondria, and the export of triglycerides as very-low-density lipoproteins to the hepatic vein [16], supporting the synthesis of very-low-density lipoprotein with methionine as the methyl donor for phosphatidylcholine synthesis. Phosphatidylcholine is the main phospholipid in ruminants, and it is critical for lipid absorption and transport, cell membrane structure, cell signaling, and very-low-density lipoprotein synthesis [17]. Choline is a precursor for phosphatidylcholine. The choline deficiency causes fatty liver, presumably due to lower levels of phosphatidylcholine biosynthesis, which impairs the export of hepatic triacylglycerols as very-low-density lipoproteins and causes hepatic steatosis [18]. It has been reported that choline is an important dietary component of Holstein, Montbeliarde, and Swedish Red crossbreed cows during the transition period as it significantly impacts the production, health, and reproduction of dairy cows [19].

Trimethyl ethanolamine, called choline [(CH_3_)3N + CH_2_CH_2_OH], is a methyl donor. It is synthesized endogenously and is a key methyl donor in mammals; its availability is important for various biological functions [8,20]. Choline is available for absorption, and more than 80% is extensively degraded by the rumen microbial population [20]. Sharma and Erdman [21] reported that the percentage of dietary choline and synthetic choline degradation in the rumen varies according to the type of feed. Choline must be administered in the rumen-protected choline (RPC) form. RPC is choline chloride protected by a fatty acid matrix layer. Rumen microbes cannot digest fatty acid; consequently, RPC reaches the small intestine, the enzymes there break down the fatty acid layer, and choline chloride becomes free for absorption. Naturally, choline chloride is less degradable than synthetic-occurring choline in the feed [22].

Synthetic drugs can pose serious problems since they are toxic and costly. By contrast, herbal medicines are relatively nontoxic, cheaper, and eco-friendly [23]. By spreading organic farming, herbal preparations have greater prospects. RPC products can be replaced by plant feed additives containing phosphatidylcholine, which shows natural resistance to ruminal degradation [19]. For example, pure phosphatidylcholine preparations were isolated from spinach leaf chloroplasts, spinach leaf microsomes, and cauliflower inflorescence. Herbal choline preparations are more acceptable than synthetic ones in modern cow nutrition strategies [24].

According to the literature data, choline’s effect on cows’ health and production is controversial. Recent research indicates that choline supplementation improves dairy cows’ lactation performance and fertility [15]. Moreover, protected choline added to their rations leads to the release of more methionine for milk protein synthesis and positively affects milk protein levels [1]. Choline reduces fatty liver conditions, and due to its presence in biological structures, it helps in reproductive tract recovery after the post-calving period. Changes in progesterone (P4) concentration are useful for predicting reproductive function recovery [25]. Researchers proved that rumen-protected choline positively affects the days leading to first observed heat, service period (days open), number of services, conception rate, and pregnancy rate [26]. Other research studies show a lower incidence of metritis, endometritis, pyometra, and retention of the placenta [27]. On the other hand, choline supplementation does not affect hyperketonemia [28]. BHB concentration is one of the markers for hyperketonemia diagnosis in dairy cows and BHB is one of the predominant circulating ketone bodies in ruminants [29]. The measurement of BHB is a useful diagnostic tool for predicting health performance in dairy cows [30].

The objective of our study was to evaluate the effects of organic herbal preparations containing choline analogs for changes in milk BHB and P4, which are associated with the reproduction and production performance of dairy cows.

## 2. Materials and Methods

### 2.1. Animals

Our study was conducted at a dairy farm (located by WGS 55.107844, 24.223197) from 1 April 2020 to 1 November 2021, where cows were housed in a cold, loose housing system with automatic cattle drinkers, no litter, and bearings lined with rubber mats.

A total of 120 multiparous (mean of parity = 3.6; SD = 1.5) healthy (without signs of lameness, mastitis, or digestive disorders) Lithuanian Black and White dairy cows with body condition scores ranging from 3.75 to 4.0 (5-point scale [8]) and similar milk yields in their previous lactation (9000–10,000 kg/lactation) were used in our study. Cows were selected 20–22 days before calving and randomly assigned to two similar groups—control (CONT, *n* = 60) and experimental (CHOL, *n* = 60)—according to body condition, live weight, parity, and milk yield of their previous lactation. They remained in these groups until 64 DIM. All cows were fed a TMR diet NO. 1 (Diet1) until calving and with diet NO. 2 (Diet2) post-calving until 64 DIM. The CHOL group’s Diet1 and Diet2 were supplemented with 10 g of a rumen-protected choline (RPC) source (Herb-AllTM, Switzerland, Mels) from 20 days pre-calving to 64 days post-calving. The rations were calculated using the HYBRIMIN 2008^®^ Fütterungs software (Hessisch Oldendorf, Germany) for dry cows (Diet1) and fresh cows (Diet2). Diet1 was formulated to meet the requirements of a 650 kg Lithuanian Black and White cow during the transition period (20 days pre-calving). The ration’s chemical composition was 11.8 kg (DM) (of the total ration), with crude protein accounting for 1.6 kg (DM), crude fiber for 2.7 kg (DM), fiber carbohydrates for 2.3 kg (DM), and non-fiber carbohydrates for 2.2 kg (DM), and the total energy for lactation was (NEL) 6.0 MJ/kg DM. The total mixed ration consisted of corn silage 26.1%, wheat straw 14.9%, grass silage 14.9%, sugar beet pulp silage 11.2%, grain concentrate mash 14.01%, and water. Diet2 was formulated to meet the requirements of a 650 kg Lithuanian Black and White cow producing 37 kg/d of milk (4.2 % fat, 3.4 % protein). The ration’s chemical composition was 24.0 kg (DM) (of the total ration), with crude protein accounting for 4.1 kg (DM), crude fiber for 3.4 kg (DM), fiber carbohydrates for 2.5 kg (DM), and non-fiber carbohydrates for 7.3 kg (DM), and the total energy for lactation was (NEL) 7.37 MJ/kg DM. The total mixed ration consisted of corn silage 33.9%, grass silage 20.9%, sugar beet pulp silage 17.4%, grain concentrate mash 13.0%, wheat straw 1.04%, molasses 1.7 %, and water. Feeding was conducted mechanically using a Keenan Klassik 100 divider mixer (Keenan, Ireland, Boxford) with electronic scales.

Dry cows were fed once daily at about 07:00 a.m., and cows post-calving were fed twice daily at 05:00–06:00 a.m. and p.m. according to the group diet (CONT or CHOL). Cows were milked automatically using the DeLaval milking robot (DeLaval Inc., Tumba, Sweden) twice daily at 04:00 a.m. and 4:00 p.m.

### 2.2. Measurement

The β-hydroxybutyrate (BHB) and progesterone (P4) concentrations in cow’s milk were determined using a fully automated real-time analyzer, the Herd Navigator (Lattec I/S. Hillerød, Denmark), combined with the milking robot. The milking robot automatically obtained a sample of a couple of milliliters of milk from each CONT and CHOL group at 5 DIM for BHB and 21 DIM for P4 determination. The subsequent milk samples were obtained according to the system’s algorithm on average every 2.2 ± 1.9 days. The analyzer measured the concentration of BHB and P4 in the milk using a biosensor and sent the BHB and P4 values to the management system DelPro 4.2 (DeLaval International, Tumba, Sweden).

Information on the milk yield of each cow was accumulated by the management system DelPro 4.2. (MI, USA) The milk composition was evaluated every 30 days (control milking), starting from 10 to 15 days post-calving. The analysis of milk fat, protein, and SCC content was performed using the infrared mid-range meter LactoScope FTIR (FT 1.0. 2001; Delta Instruments, Drachten, The Netherlands), where the number of milk constituents was determined from the absorbed energy. The somatic cell count in milk was determined using Somascope (CA–3A4, 2004; Delta Instruments, Drachten, The Netherlands) equipment and the principle of flow cytometry. The experimental milk was first mixed with a staining solution, after which the mixture entered the part of the device where it was illuminated with ultraviolet light, activating the fluorescent molecules and causing each stained cell to glow. A computer program registered the received signals and somatic cell count per milliliter of milk.

Data on the insemination rate (the insemination times performed until cows from the group became pregnant divided by the number of pregnant cows), or the period between calving and the first insemination time, were obtained from the DelPro 4.2. (MI, USA) farm management platform.

### 2.3. Statistical Analysis

Experimental research data were analyzed using statistical software SPSS (version 25, SPSS Inc., Chicago, IL, USA). The data were presented using descriptive statistics and normal distribution analysis methods, such as the Kolmogorov–Smirnov test. The results were produced as the mean ± standard error of the mean (M ± SE). ANOVA was used to analyze the differences between the mean values of normally distributed variables. The differences were considered significant when *p* ≤ 0.05.

## 3. Results

### 3.1. Change of β-Hydroxybutyrate Concentration at Different Intervals of DIM

The mean BHB concentration was not statistically significant during the interval of 5–64 DIM. The mean BHB concentration was 5.92% lower in the CHOL group than in the CONT group with *p* > 0.05 (Figure 1).

### 3.2. Change of Progesterone Concentration at Different DIM Intervals and Reproduction Properties

The average progesterone (P4) concentration at 21–24 DIM was 45.65% higher in the CHOL group than in the CONT group. Despite noticeable differences, the difference was not statistically significant. The P4 concentration of the CHOL group was 7.63% higher than the CONT group at 21–64 DIM with *p* > 0.05. P4 differences between the CHOL and CONT groups at different DIM intervals showed that the P4 change cycle was earlier in the CHOL group. The first peak of the P4 concentration was determined 4.81 days earlier in the CHOL group than in the CONT group of cows with *p* < 0.001 (Figure 2).

The CHOL group of cows had superior reproduction properties. The insemination rate was 11.21% lower in the CHOL group than in the CONT group, with *p* < 0.05. The interval between calvings was 4.67% shorter in the CHOL group than in the CONT group with *p* < 0.05. Furthermore, the first insemination time was 8.86% earlier in the CHOL group than in the CONT group of cows with *p* < 0.05 (Table 1).

### 3.3. Change of Milk Yield and Milk Composition at Different DIM Intervals

The milk yield of cows at different DIM intervals was statistically significant between the CHOL and CONT groups. Milk yield during the first days (5–24 DIM) of lactation was 0.08% lower in CHOL than in the CONT group at 5–20 DIM (*p* > 0.05). Subsequently, the difference in milk yield was more noticeable between the CHOL and CONT cow groups. Milk yield for the 21–64 DIM period was 7.87% higher in the CHOL group than in the CONT group with *p* < 0.001 (Figure 3).

The milk composition was different between the CHOL and CONT groups. In the period of 10–15 DIM, more fat was detected in the milk of the CHOL group (0.36%) (*p* > 0.05) and 0.58% more for the period of 40–55 DIM than in the CONT group (*p* < 0.05). The milk of the CHOL group contained less protein (0.09%) at 10–15 DIM and 0.23% more at the 40–55 DIM period than the CONT group (*p* > 0.05). Furthermore, milk from the CHOL group had less SCC (45.83%) at 10–15 DIM and 8.2% more at 40–55 DIM than the CONT group, with *p* > 0.05 (Figure 4).

## 4. Discussion

The most important findings of our research are presented in Figure 5. We found that administering a choline supplement to cows improved their reproductive cycle. The following reproductive parameters improved as progesterone reached its first peak: insemination rate, the period between calvings, the first day of insemination, increased milk yield, and milk composition.

No statistically significant differences were observed in BHB concentrations between the CHOL and CONT cow groups. In previous studies, the effect of rumen-protected choline supplementation on dairy cows’ health disorders was different from ours. Several research works claimed that supplementing RPC at post-partum did not affect the plasma concentrations of BHB [16,17]. On the other hand, Feifei Sun et al. established that feeding RPC reduced the plasma concentration of BHB [18]. Morrison et al. found that supplementing choline with vitamin B influenced the BHB concentration depending on DIM. Treatment did not affect the incidence of hyperketonemia (blood BHB ≥ 1.2 mmol/L to 3 weeks post-partum); however, this indicator was lower in the third week [31]. No significant changes in BHB concentration were related to increased productivity. Hyperketonemia is also related to liver function and triacylglyceride uptake [32]. The inability of supplemental choline to reduce hepatic triacylglycerol may have resulted from increased production performance without additional dry matter intake [33]. In general, choline supplementation may be beneficial due to favorable changes in liver function [34]. Choline’s impact on liver function can be explained by its interaction with glucose metabolism. Increased choline concentration in hepatic cell culture decreased glucose but increased cellular glycogen. During in vitro conditions, fatty acids did not affect glucose or cellular glycogen but increased pyruvate carboxylase cytosolic and mitochondrial phosphoenolpyruvate carboxykinase with choline treatment [35].

Feeding RPC during the transition period led to shorter intervals from calving to first luteal activity (CLA) and better reproduction performance parameters (first insemination time, insemination rate, and period between calvings). Our results showed that the progesterone (P4) profile during 21–64 DIM was similar in the CHOL and CONT groups (*p* > 0.05). However, we detected a P4 concentration peak earlier in the post-calving period in the CHOL group than in the CONT group. In our opinion, RPC tended to accelerate the first insemination time in 7.3 days and for 19.4 days, shortening the period between calvings in the CHOL group of cows. We also observed that the insemination rate of cows was lower in the CHOL group than in the CONT group. Similar results were reported by Zenobi et al. (2018) who found that rumen-protected choline (RPC) supplemented during the transition period improved the conception rate after insemination and also conceivably affected embryonic development [21].

According to Peterssona et al. (2006), CLA is a significant marker in early lactation for traditional fertility measurements, such as pregnancy to first insemination, the number of inseminations per service period, and the interval from first to last inseminations [36]. However, there is less information on RPC’s effect on plasma P4 concentrations in cows. Lima et al. showed that feeding cows with RPC did not affect the P4 concentration. However, E. Humer et al. reported the contradictory effects of RPC on reproduction [9]. A. Ardalan et al. and C. Furken et al. observed improved reproduction in cows supplemented with RPC, such as increased cyclicity and pregnancy rates [22,37]. Lima et al., A. Pirestani and M. Aghakhani reported controversial effects of RPC on reproduction [26,28,38]. Research has shown that choline helps cows to balance negative energy during particular reproductive periods, resulting in increased follicular development and fertility. In addition, choline deficiency causes decreased hormone production (FSH and LH) because of the necessity of choline in cell membrane structures.

Our research results showed that supplementing RPC affected cows’ milk yield and composition. RPC supplementation was 7.87% higher in CHOL compared with the CONT group during 21–64 DIM and was similar to the results established by Jayaprakash [3]. He stated that choline improves milk production in lactating dairy cows by approximately 7% compared to the control. Other studies have also shown that choline positively affects milk yield [2,33] These results can be explained by the fact that choline is used for liver protection, improves fat metabolism in the liver, and simultaneously reduces triglyceride levels. As a result, the liver produces more glucose, which is used for greater milk synthesis [37].

In our research, RPC supplementation increased the fat content in milk. Supplementation with RPC facilitates phospholipid synthesis, lipid absorption, and transport to the mammary gland, thus favoring milk fat synthesis [3,19,20,21]. In addition, SCC in both groups was within the normal range, and there were no statistically significant differences between the groups (*p* > 0.05), indicating that healthy cows participated in the study.

## 5. Conclusions

RPC supplementation for dairy cows from the second week before calving to the second week of post-calving did not affect the BHB concentration during the early post-partum period. Feeding cows RPC influenced their reproductive performances. We detected the first peak of P4 earlier, a shorter first insemination time, decreased insemination rate, and a shorter period between calvings when cows were fed RPC. RPC supplementation significantly increased milk yield and fat content, but no statistically significant differences were observed in milk protein and SCC at 60 DIM. Therefore, further studies are required to quantify the impact of rumen-protected choline supplementation on dairy cows’ health status and reproduction.

## Figures and Tables

**Figure 1 animals-12-01807-f001:**
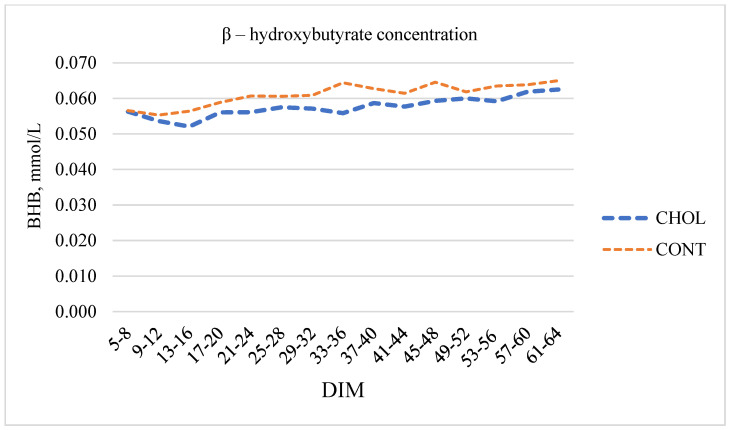
Changes in β–hydroxybutyrate concentration in milk of the choline and control groups at different DIM intervals.

**Figure 2 animals-12-01807-f002:**
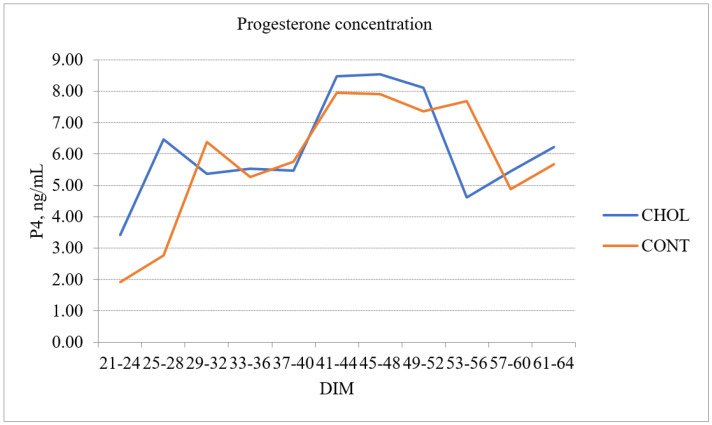
Changes in progesterone concentration in milk of the choline and control groups at different DIM intervals.

**Figure 3 animals-12-01807-f003:**
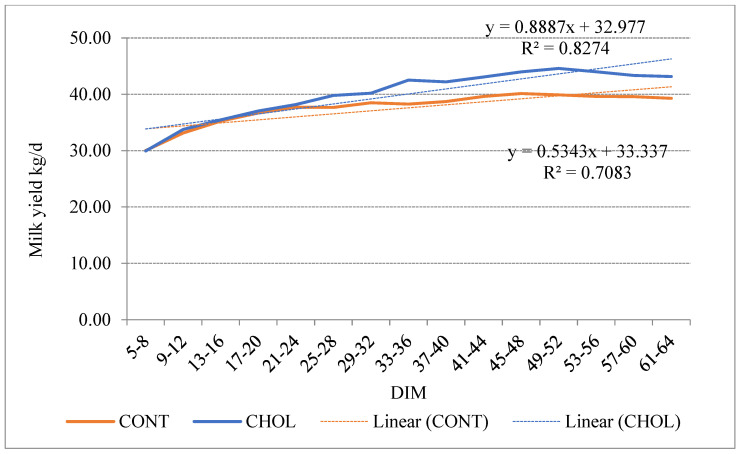
Changes in cows’ milk yield for the CHOL and CONT groups at different DIM intervals.

**Figure 4 animals-12-01807-f004:**
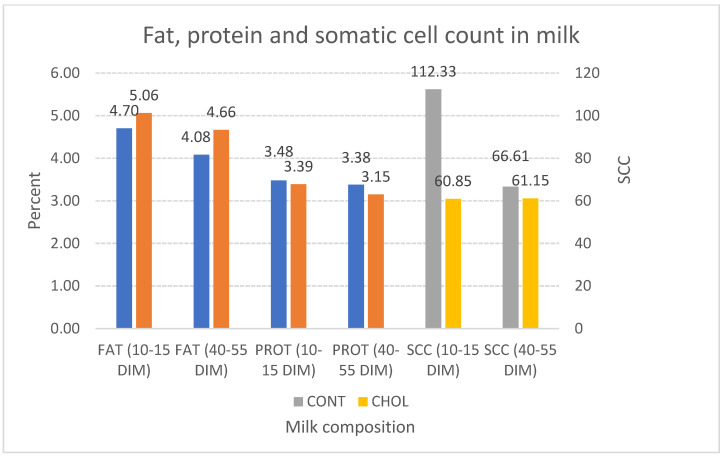
Milk composition of the CHOL and CONT groups at 10–15 and 40–55 DIM.

**Figure 5 animals-12-01807-f005:**
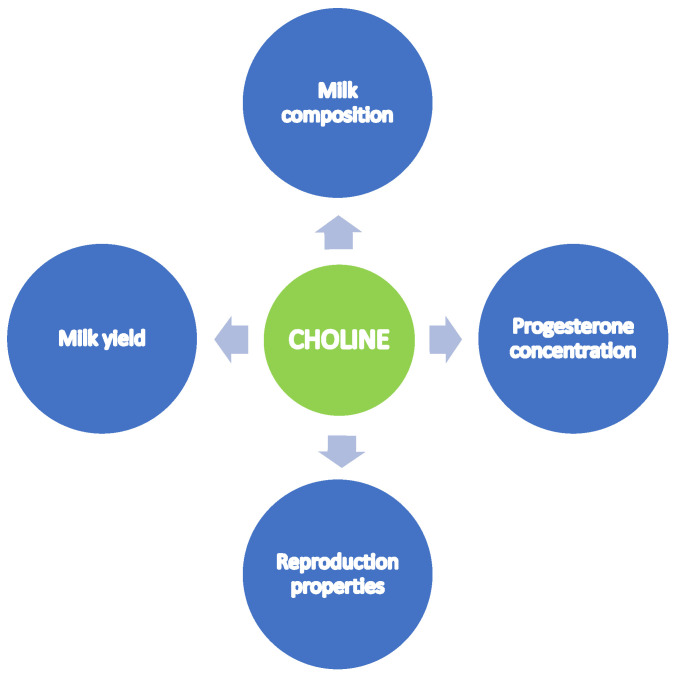
Positive effect of choline supplementation.

**Table 1 animals-12-01807-t001:** Reproduction traits of cows in the CHOL and CONT groups.

Group	Insemination Rate		Interval between Calvings		First Insemination Time (in Days)	
	M	SEM	M	SEM	M	SEM
CONT	2.14	0.28	415.94	12.00	82.49	2.45
CHOL	1.90	0.25	396.52	5.7	75.18	2.11

## Data Availability

The research was conducted on a private farm (located by World Geodetic System (55.107844, 24.223197)) based on private data from that farm.

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
