# Peer review of "The Effect of Supplementation of Rumen-Protected Choline on Reproductive and Productive Performances of Dairy Cows"

_animals, 2022, doi:10.3390/ani12141807_

Round 1

Reviewer 1 Report

This is an interesting study on the improvement of choline cow reproductive and productive parameters. These results are important for cow breeders. Some improvements are suggested before publication

 -“develop metabolic and/or infectious diseases during postpartum period”- give more details

- “the influx of NEFA into the liver”-give more details and information on NEFA

-“Therefore, choline is an important component of the diet of dairy cow’s during the transition period, as it has a significant impact on production, health and reproduction [14].”-give cow breed

It will be nice to have a schematic diagram showing the most important discoveries of your research

Author Response

Thank You for your time and comments.

Reviewer 2 Report

The paper describes an experiment to study the lactational and reproductive effects of rumen protected choline supplementation in Lithuanian Black transitional dairy cows.  The paper builds on previously published work elsewhere on the role and potential benefits of choline supplementation but provides more specific localised evidence under Lithuanian conditions.  Thus the paper is not highly original but provides practical evidence under local conditions.  The paper is publishable with the provision of substantially more information.

Could the authors please provide further information and comment as follows:

1.  Were the control and CHOL groups balanced for body condition, live weight, parity, previous lactation yield?

2.  Please provide more detail on the technical specifications of the choline supplement.  Is there good evidence to show that this supplement results in higher plasma levels of choline?  Did they measure choline levels in the blood?

3.  How were the milk assays for BHB and P4 validated?

4. Please define 'insemination rate' more precisely

5.  Given the marginal effect on milk yield and fertility parameters, what is the likely cost-effectiveness of choline supplementation? 

6.  Can the authors please state what they see as the practical value of the findings from this paper.

Author Response

Thank You for your time and comments.

Round 2

Reviewer 2 Report

I think that with some further editing of the English language this paper can now be published.

Author Response

Hi, thank you for your advice, we edited manuscript   :)
